# Multicenter Study of Whole Breast Stiffness Imaging by Ultrasound Tomography (SoftVue) for Characterization of Breast Tissues and Masses

**DOI:** 10.3390/jcm10235528

**Published:** 2021-11-25

**Authors:** Peter J. Littrup, Nebojsa Duric, Mark Sak, Cuiping Li, Olivier Roy, Rachel F. Brem, Linda H. Larsen, Mary Yamashita

**Affiliations:** 1Department of Radiology, Karmanos Cancer Institute, Detroit, MI 48201, USA; 2Department of Radiology, Wayne State University, Detroit, MI 48201, USA; 3Delphinus Medical Technologies Inc., Novi, MI 48374, USA; nebojsa_duric@urmc.rochester.edu (N.D.); msak@delphinusmt.com (M.S.); cli@delphinusmt.com (C.L.); oroy@delphinusmt.com (O.R.); 4Department of Radiology, University of Rochester, Rochester, NY 14642, USA; 5Department of Radiology, The George Washington Cancer Center, George Washington University, Washington, DC 20037, USA; rbrem@mfa.gwu.edu; 6Department of Radiology, Norris Cancer Center and Hospital, University of Southern California, Los Angeles, CA 90033, USA; Linda.Larson@med.usc.edu (L.H.L.); Mary.Yamashita@med.usc.edu (M.Y.)

**Keywords:** breast cancer, ultrasound tomography, automated breast ultrasound, stiffness, fibroglandular tissue composition, fibroadenoma

## Abstract

We evaluated whole breast stiffness imaging by SoftVue ultrasound tomography (UST), extracted from the bulk modulus, to volumetrically map differences in breast tissues and masses. A total 206 women with either palpable or mammographically/sonographically visible masses underwent UST scanning prior to biopsy as part of a prospective, HIPAA-compliant multicenter cohort study. The volumetric data sets comprised 298 masses (78 cancers, 105 fibroadenomas, 91 cysts and 24 other benign) in 239 breasts. All breast tissues were segmented into six categories, using sound speed to separate fat from fibroglandular tissues, and then subgrouped by stiffness into soft, intermediate and hard components. Ninety percent of women had mammographically dense breasts but only 11.2% of their total breast volume showed hard components while 69% of fibroglandular tissues were softer. All smaller masses (<1.5 cm) showed a greater percentage of hard components than their corresponding larger masses (*p* < 0.001). Cancers had significantly greater mean stiffness indices and lower mean homogeneity of stiffness than benign masses (*p* < 0.05). SoftVue stiffness imaging demonstrated small stiff masses, mainly due to cancers, amongst predominantly soft breast tissues. Quantitative stiffness mapping of the whole breast and underlying masses may have implications for screening of women with dense breasts, cancer risk evaluations, chemoprevention and treatment monitoring.

## 1. Introduction

Elasticity assessment became part of the 5th edition of Breast Imaging Reporting and Data System (BI-RADS) for handheld ultrasound (US) in 2013, created under associated features of masses as soft, intermediate or hard [1]. However, its adoption in routine clinical care has been limited. Lack of standardization for quantitating tissue stiffness can be attributed, in part, to differing physics approaches (i.e., shear-wave vs. strain, or Young’s moduli) and vendor-specific implementations (e.g., red to blue vs. blue to red stiffness color scales) [2,3]. We evaluated a marker of bulk modulus obtained by ultrasound tomography (UST) that displays relative stiffness and can be volumetrically quantified for both whole breast and mass-specific analyses. The bulk modulus describes the strain response of a body to stress involving change in volume without change of shape, which can be thought of as compressibility or stiffness [2,3]. Young’s modulus, on the other hand, is a measure of the ability of tissue to withstand changes in length when under lengthwise tension or compression, sometimes referred to as the modulus of elasticity used in US strain elastography (SE). The shear modulus relates to the strain response of a body to shear or torsional stress, involving change of shape without change of volume, used in shear wave elastography (SWE). In other words, bulk modulus is the ratio of volumetric stress to volumetric strain, Young’s modulus is the ratio of tensile stress to tensile strain, and shear modulus is the ratio of shear stress to shear strain. These differences drive different technical solutions for imaging each type of tissue modulus, each with its own proprietary method for displaying these properties, which may also limit adoption.

Another barrier to clinical adoption of elasticity assessments is the limited field of view using handheld US (HHUS) and the associated operator dependence that limits workflow. Moreover, stiffness parameters have not been included in dense breast screening efforts by HHUS. While current automated breast US (ABUS) effectively images the whole breast [4] and can be used in screening, it does not provide stiffness imaging. The volumetric bulk modulus may thus provide a whole breast imaging solution, providing tissue stiffness measurements that can be standardized between patients and tissues. Therefore, visualization of UST stiffness throughout the whole breast may help address practical issues for consistent scanning of stiffness parameters, as well as their subsequent diagnostic interpretations.

UST is an emerging form of ABUS that follows many years of research and development by a variety of groups [5,6,7,8,9,10,11,12,13]. Several clinical evaluations of the SoftVue UST system (Delphinus Medical Technologies, Novi, MI, USA) have been reported [5,6,7,14,15,16,17,18], but this is the first that details stiffness outcomes using a proprietary version of bulk modulus that depicts relative tissue stiffness from the entire visualized breast, including underlying masses. We present these multicenter trial results of whole-breast stiffness imaging by SoftVue, with the goal of characterizing fat, fibroglandular tissue, benign and malignant masses. The diagnostic implications of characterizing whole breast stiffness are discussed in the context of common breast masses and future research paths noted.

## 2. Material and Methods

### 2.1. Subjects and Masses

Data were obtained from scans of patients that were recruited to the diagnostic arm of a HIPAA compliant, multi-arm, multicenter trial of SoftVue UST [Clinicaltrials.gov–NCT#02977247: Delphinus SoftVue Prospective Case Collection-ARM 2 (SV PCC ARM2)]. Data from the other arm of the multi-center study using UST for dense breast screening as an adjunct to mammography (i.e., NCT03257839: Delphinus SoftVue Prospective Case Collection-ARM 1 (SV PCC ARM1) are not reported here and could not be accessed while under FDA review of Pre-Market Approval (PMA). Patients in ARM2 were eligible to receive SoftVue imaging as part of their clinical visit for evaluation of a palpable or mammographic abnormality. Informed consent was thus obtained from all women within this observational cohort study whereby their main inclusion criterion was their willingness to participate with a SoftVue scan during their clinical visit. Notable exclusion criteria were age <18 years, body weight >350 pounds (i.e., SoftVue scanning table projected limit), inability to give informed consent, inability to lie prone on the UST table, and any open sores or wounds on the breast precluding immersion into the UST water bath. Water within the SoftVue scanning tank is exchanged between patients and sanitized with ProTex (Parker laboratories Inc., Fairfield, NJ, USA). For this study, all patients were included between UST scan dates 4/2017–10/2018 for this consecutive data set, using the same version of the SoftVue unit and associated reconstruction algorithms across all centers of the trial.

All identified masses were biopsy-confirmed by subsequent or prior histology, unless considered as a characteristic cyst by hand-held ultrasound criteria. All complicated cysts underwent aspiration with cytologic confirmation. Some women had more than one mass in each or both breasts. Masses were grouped into the main categories of cancer, fibroadenoma, cysts and other benign (i.e., fibrosis, etc.), then separated according to size (≤1.5 cm and >1.5 cm diameter), which also matched the set-point of 1.5 cm for post-processing using a high pass spatial filter (see also Section 2.4: Mass Stiffness Distributions). Masses 5–15 mm are also a commonly targeted size range for breast cancer screening. No apparent mass calcifications were noted on mammography or handheld ultrasound that could have contributed to, or significantly altered stiffness values.

### 2.2. Equipment–SoftVue Ultrasound Tomography (UST) and Stiffness Imaging

SoftVue has been FDA-cleared as comparable to previously existing breast US and elastography technology (i.e., 510K numbers: K123209 and K142517) and just recently received PMA as an adjunct for dense breast screening (PMA# P200040; https://www.accessdata.fda.gov/scripts/cdrh/cfdocs/cfpma/pma.cfm?id=P200040 (accessed on 21 September 2021). A SoftVue scan is operator-independent and covers the entire volume of the breast, up to but not including the axilla. The patient lies prone on a table that houses a water bath in which the breast is pendant during scanning. A ring-shaped sensor surrounds the breast inside the water bath and scans the whole breast from nipple to chest wall (i.e., nipple image labeled #1, then increasing toward the chest wall) in approximately two minutes, providing a stack of 2.5 mm thick coronal images (Figure 1). SoftVue’s operating characteristics (Table 1) include a frequency range of 1–3 MHz and a spatial resolution of 0.75 mm in the coronal acquisition plane and 2.5 mm out of plane. The coronal image stacks are co-registered for their different presentations, providing clinical image stacks of reflection and two stacks of transmission images consisting of sound speed and stiffness fusion, the latter of which incorporates attenuation (Figure 1). Thumbnail axial and sagittal reconstructions provide 3D localization, along with the sequential coronal image review. Furthermore, patient position matches the appropriate clock position and an external calibrated encoder provides the anterior-posterior (AP) distance relative to the nipple. For the purposes of this study, it is the stiffness and sound speed image stacks that represent the volume of the breast to be analyzed and interpreted.

The transmitted signal that is used for all SoftVue imaging, is a longitudinal wave pulse. Since the bulk modulus represents material resistance to compression from a longitudinal wave [2,3,19,20,21,22], it is theoretically possible to image the bulk modulus. The advantage of such a method is that there is no need for separate excitation of the tissue. The information can be extracted from the existing imaging data. Since the pulse shape and strength can be varied, the potential exists for a larger dynamic range with greater tissue differentiation by bulk modulus than current strain or shear wave elastographic methods [23]. The derivation of a bulk modulus surrogate for SoftVue that displays numerical pixel values of relative stiffness has been described [7,15]. Briefly, a hard lesion, is also a dense lesion with high sound speed, whereas a non-solid soft lesion is characterized by low levels of attenuation to longitudinal waves. Therefore, combining attenuation and sound speed into a single stiffness parameter, or surrogate bulk modulus, yields low values of stiffness (e.g., blue) for non-solid lesions such as cysts and high stiffness (e.g., red) for solid masses, like cancers. As noted in the accompanying SoftVue BI-RADS paper in this issue, evaluating stiffness data is the last step in characterizing a mass after it has been detected by either wafer, reflection or sound speed imaging.

UST stiffness may be referred to as a surrogate of the bulk modulus since no calibration was used to tie it to bulk modulus via an external standard. Hence, no absolute measurement of the value of the derived bulk modulus was possible. Nevertheless, since the parameters used to define the surrogate (i.e., sound speed and attenuation) are tied to an external standard, the pixel values of the derived surrogate can be used as a consistent reference for comparing relative differences in the bulk modulus between patients. Furthermore, during UST image processing, pixel values are calculated and displayed on a color scale, designed to map the full range of stiffness values found in a given breast, optimized for relative stiffness differences in that breast. In this study, we chose a range of colors that are segmented into groups as noted below, emphasizing that the stiffness parameter is displayed as quantified pixel values that can be seen on a relative color scale and reproduced from patient to patient.

### 2.3. Whole Breast Tissue Stiffness Distributions

Sound speed images use quantitative pixel values (i.e., m/s) that can be separated into the tissue categories of fibroglandular (i.e., mammographically “dense”) or fatty by the use of K-means clustering techniques that classified all pixels as either white or black (i.e., fibroglandular or fat, respectively) [6,12,15,16,17,18]. Applying this same segmentation procedure to the quantified pixel values within the stiffness images allowed for the creation of masks with three relevant component categories of soft, intermediate and hard in analogy to US-BIRADS descriptors [1]. Six overall whole breast tissue components were defined by the intersection of the two sound speed masks with the three stiffness masks (i.e., a region that appears hard on the stiffness image, and fibroglandular on the sound speed image, was classified as “hard fibroglandular”). The volume of each of the six regions can be measured by counting the voxels in each mask with the sum of all six regions corresponding to the total volume of the breast. While a MIM workstation (MIM Software Inc., Cleveland, OH, USA) was used to visually display the different images, all analyses were performed using ImageJ.

Correlations between sound speed and stiffness values were evaluated to assess their independence, especially given that stiffness is derived in part from incorporating sound speed values. The total volume of each tissue component was determined for each patient and then averaged across the patient cohort. Similarly, the percentage of each such component was also determined.

### 2.4. Mass Stiffness Distributions

Mass boundaries were hand-traced by a radiologist with extensive UST experience (PL) and over 20 years as a MQSA-certified breast radiologist, using MIM software to generate regions of interest (ROIs) surrounding all margins of detected masses. Mass margin contours were traced on the single most representative coronal image using a combination of sound speed and reflection image stacks. Careful note was made of these underlying margins to clearly separate tumor from peritumoral regions which have also been documented in quantitative analyses of mass locations [24]. Minor errors in tumor contours were thus minimized for this single user by selecting the most representative single coronal image as further work progresses toward automated margin detection of mass volumes. The whole breast masks that were generated above were then intersected with the hand-traced ROIs to create the percentage tissue distributions within the optimally traced surface area of each mass. Since radiologists sequentially view these individual high resolution coronal images, single slice analyses of mostly small masses are relevant and representative of future volumetrics.

The average stiffness index of each mass ROI was defined on a scale of 0 to 1, based on the pixel values of stiffness. As noted from segmentation using K means clustering, the pixel values were distributed to the three stiffness levels that roughly correspond to the range of colors on the stiffness image (i.e., soft = blue-black; intermediate = green-yellow; hard = orange-red). Each mass thus also had an associated color pattern, in addition to their average stiffness index. The corresponding stiffness distributions were then determined for each major mass types of cancers, fibroadenomas and cysts, across the patient cohort to assess any differences. The stiffness properties of the masses were compared with the corresponding whole-breast values to identify any associations of mass type with the six tissue components.

Spatial filtering is a relatively common post-processing step that can emphasize or deemphasize structures based on their size [25]. A high pass spatial filter (i.e., suppressing spatial scales >1.5 cm) was applied to all images in order to emphasize masses while deemphasizing adjacent fibroglandular tissue that could mask some of the mass properties. The impact of this spatial filter on mass stiffness in relation to mass size and type was evaluated for its initial impact and potential use with future post-processing options.

### 2.5. Statistical Analyses

For this observational cohort study, descriptive statistics were utilized. The ability to statistically differentiate masses using stiffness parameters was assessed using the *t*-test. Mass stiffness indices, derived from average pixel values, were also assessed for texture differences (e.g., color patterns) by testing the single higher-order statistical feature of homogeneity. This provided some insight to future uses of more complete radiomics (i.e., beyond the encoding scheme of gray-level co-occurrence matrices (GLCOM)] [26]. Frequency differences were determined by the standard chi-squared test. Significance was based on a *p* value < 0.05.

## 3. Results

### 3.1. Subjects and Masses

The average age for participants in this study was 48.9 years (standard deviation = 11.6 years, range 18–82 years). The retrieved data sets represent 239 individual breasts from 206 diagnostic patients with 298 masses, including cancer and benign masses, as noted in Table 2. There were 1.4 masses per woman (298/206) or 1.2 (239/206) masses per SoftVue scanned breast. The average total breast volume was 737 mL and the average tumor volume was 1.1 mL. Over 90% of patients had heterogeneously or extremely dense breasts by mammography (i.e., N = 133 (64.6%), or N = 55 (26.7%), respectively). Patients with suspicious masses were also included from women with scattered breast density (N = 18, or 8.7%) as part of the SV PCC ARM2, but no masses were encountered in women with nearly all fat breast density. This clinical cohort of woman with breast masses is likely skewed toward the higher mammographic breast densities because the other screening arm of the multi-center study was aimed at assessing UST performance in women with dense breasts.

### 3.2. Whole Breast Stiffness

The association of stiffness vs sound speed values was negligible (correlation coefficient of 0.0048) suggesting that the stiffness property of tissues is independent of its sound speed component.

The averages associated with the six tissue components, totaling 737 cc, are summarized in Table 3. On average, 29% of the breast consisted of fibroglandular tissue vs. 71% fatty tissue. As expected, 97% (507/523 cc) of the fatty tissue was composed of soft or intermediate components. On average, 31% (66/214 cc) of the fibroglandular tissue was classified as hard, while 29% (61/214 cc) was soft and the remaining 40% intermediate. A histogram of the six whole-breast tissue categories, according to the types of masses they harbor, is presented in Figure 2.

Images of sound speed, reflection and stiffness are shown in Figure 3 to illustrate the typical spatial distribution of the tissue components relative to breast anatomy. Fibroglandular tissue is represented by the brighter regions in the sound speed and reflection images. The coloration of stiffness images does not appear proportionate to the relative brightness within sound speed or reflection images, consistent with the extremely low correlation coefficient of stiffness vs. sound speed values.

### 3.3. Mass Stiffness Distributions

Examples of filtered and unfiltered images of a mammographically occult cancer are shown in Figure 4. The default unfiltered stiffness image shows partial obscuration of the underlying mass by the adjacent hard parenchyma (Figure 4c). The small cancer is better defined in the filtered stiffness image (i.e., non-standard Stiffness Fusion version; Figure 4d), due to partial suppression of the adjacent fibroglandular tissue.

Unfiltered and filtered stiffness distributions were separated according to mass size and type, as shown in Figure 5. In general, the smaller masses (i.e., ≤1.5 cm) had a significantly greater stiffness compared to the larger masses (i.e., >1.5 cm), regardless of tumor type or filtering option. Conversely, larger masses were significantly softer (chi-squared; *p* = 0.001). For the filtered images, small cancers were stiffer than small fibroadenomas (*t* test, *p* = 0.001). Only smaller cancers were significantly altered by spatial filtering, increasing their hard component by 23.1%, from pixel averages of 61.5% and 30.8% for unfiltered hard and intermediate components, to 84.6% and 11.5%, respectively (*p* < 0.001).

Quantitative stiffness values of large and small masses, as displayed by the unfiltered and spatially filtered algorithms, are shown in Table 4. The filtered rendering produced significantly greater discrimination of smaller cancers from fibroadenomas (i.e., *p* = 0.00036 versus *p* = 0.080; bold in upper Table 4). Conversely, the unfiltered stiffness images better separated the larger cancers from fibroadenomas (*p* = 0.037 versus *p* = 0.127; bold in lower Table 4). Stiffness indices and homogeneity texture differences between the mass types were significant for both filtered and unfiltered stiffness images, respectively (*p* = 0.035).

Additional examples of spatially filtered stiffness images are shown in Figure 6, using magnified cropped views of both smaller and larger cysts, fibroadenomas and cancers. Resultant stiffness patterns show qualitative differences in mass appearances, which is also complicated by their interactive appearances while scrolling through the coronal Stiffness Fusion image stack. Namely, true underlying masses on the background Reflection portion of the Stiffness Fusion image show colors that appear to track within the mass margins, or “sticking” to the mass (e.g., note the intermediate green colors of the larger fibroadenoma (Figure 6B, bottom) conforming to its mass boundaries). Conversely, adjacent normal parenchyma usually has its more amorphous colors “flow” from image to image, yet hard fibroglandular tissue can still partially obscure a cancer as an Figure 4 (see also accompanying SoftVue BI-RADS article in this issue for more detailed mass confirmation and characterization steps).

Considering benign masses first, simple cysts had a soft appearance (blue-black color) with little or no internal stiffness, regardless of size (Figure 6a). Smaller cysts containing stiffer components were commonly associated with complicated cysts (i.e., by standard US approaches) and underwent aspiration/biopsy. Fibroadenomas had either homogeneous or mildly heterogeneous internal stiffness (Figure 6b), compatible with the volumetric distributions noted in Figure 5 and the more uniform blending of the stiffness components. Within the fourth histologic category of “other benign” (Table 2; N = 24), three larger masses showed a softer pattern similar to cysts. Conversely, the smaller other benign category (i.e., N = 21) had predominantly underlying fibrosis (i.e., biopsy report descriptions) with stiffness similar to cancers, thus representing false positives that underwent confirmatory biopsy. Nonetheless, all these hard fibrotic masses represented only 13% (21/161) of solid masses <1.5 cm (Table 2) and were still concordant on standard imaging, not requiring re-biopsy or excision.

Cancers displayed visual characteristics in Figure 6 that also showed size-related quantitative improvement in mass differentiation after limited post-processing by spatial filtering (Table 4). As noted, the spatial filtered images of smaller cancers showed 23% increased percentage of the hard component (Figure 5c and Figure 6c-top), whereas larger cancers showed ~15% decreased hard component (Figure 5b and component (Figure 6c-bottom). Qualitatively, smaller cancers often had their hard component located centrally and appeared more “filled-in” after spatial filtering (Figure 6c-top row), whereas larger cancers were mostly soft and had their decreased hard components often residing more within its residual rim (Figure 6c-bottom row). Smaller cancers also had irregular margins with less contrast on reflection (i.e., intermediate or gray), corresponding to conventional US terminology of isoechoic, rather than the darker appearance of benign masses and larger cancers. The powerful BI-RADS parameters of mass shape and margins using reflection and sound speed image stacks also convey the sequential importance of using stiffness fusion for mass characterization as the last step after confirmation of an underlying mass by the other image stacks. Finally, too few cancer sub-types (Table 2) were available for significant analyses.

## 4. Discussion

To the best of our knowledge, this study represents the first assessment of whole breast stiffness by ultrasound tomography, which combined the transmission properties of attenuation and sound speed to create a unique surrogate of bulk modulus. SoftVue stiffness imaging in the coronal plane can be volumetrically assessed and compared throughout the breast and between patients, including underlying masses that are readily localized in volumetric formats and/or the thumbnail images in axial and sagittal planes. Other investigators using FDA-cleared UST (i.e., QT Ultrasound LLC; Novato, CA, USA) have not employed stiffness or attenuation, limiting their clinical evaluations to the use of reflection and sound speed for characterizing breast density [8,10] and differentiating cysts from solid masses [9,11]. Applying quantitative UST stiffness parameters to whole breast and underlying masses for diagnostic tissue characterization may thus provide a framework for further understanding of their qualitative appearances as SoftVue transitions to dense breast screening after its recent PMA, as well as automated detection and characterization.

In related work, direct stiffness measurements of resected breast specimens have been described as a tissue spectrum, with progressive increases in overall mass stiffness, extending from the softest benign solid tissues to the hardest types of different cancers [27]. Surgical resection is primarily done only for solid suspicious masses, such that cysts were not included in their analyses. Similarly, SoftVue stiffness evaluations throughout the breast and underlying masses demonstrated progressively stiffer appearances, from benign breast tissues to benign and malignant tissues, which now also includes cysts. Stiffness imaging using a bulk modulus surrogate was undertaken at relatively high resolution and may also avoid some of the artifactual aspects of cysts encountered with current elastography [3].

SoftVue demonstrated the ability to achieve 0.75 mm resolution in the coronal plane despite the low operating frequency of 1–3 MHz. In conventional US approaches, images are produced by assuming that acoustic waves travel as rays. Refraction and diffraction are not taken into account. Consequently, the resolution limit of *λ*/2 (set by diffraction theory, where *λ* is the wavelength of the acoustic wave) cannot be achieved and is typically > λ. With UST, data are amenable to a wave equation solution which accounts for refractive and diffractive effects and allows the reconstruction algorithm to approach the theoretical limit of *λ*/2. For example, the speed of sound algorithm utilizes the portion of the pulse spectrum near 1 MHz which corresponds to a wavelength of 1.5 mm and, therefore, a *λ*/2 of 0.75 mm which is the coronal plane resolution of the SoftVue system. At 1 MHz, a conventional system would have a resolution of several millimeters. SoftVue’s stiffness fusion imaging thus provided submillimeter resolution of stiffness patterns within common breast masses throughout the breast that can be used by current breast radiologists assessing qualitative visible differences, as well as quantitative tissue analyses that help substantiate different mass appearances.

### 4.1. Whole Breast Stiffness

As expected, fatty tissue was found to be universally soft while 69% of fibroglandular tissues were also relatively soft (Table 3). The ~2% fat volume having a hard component could relate to focal high collagen content, or fibrosis, but may in part be artifactual because K-means clustering can lead to inadvertent inclusion of fatty pixels near boundaries adjacent to fibroglandular tissues. Cancerous masses were also associated with breast volumes that had their largest percentage as fat, but may in part relate to greater inclusion of cancer patients with scattered, or more fatty, breast density. Further work is needed in assessing the role of obesity and/or the potent endocrine contributions that spur cancer initiation at the fat–glandular interface, which have been noted on breast MR [28] and UST [24].

From a simple visualization perspective, the average mass size of 1.1 mL and average breast volume of 737 mL suggest that visualization of underlying masses is a volumetric balance of relative stiffness. Namely, small cancers are mostly stiff and could either be readily seen amongst a predominantly soft background, since 9% (66 mL) of hard fibroglandular tissue would be scattered throughout that volume. Or, a cancer could be potentially obscured if it was also partially embedded in a larger grouping of hard fibroglandular tissue, such as in Figure 4. It should be noted that spatial filtering to suppress these larger groupings of fibroglandular tissue were not evaluated for whole breast stiffness since stiffness fusion applies only to mass characterization after confirmation of an underlying mass, so far. For example, applying a spatial filter or other post-processing to whole breast evaluation raises questions of additional false positives that have yet to be reviewed for future screening.

### 4.2. Mass Stiffness Distributions

The unfiltered properties of the small mass group yielded statistically significant discrimination of cysts from cancers and cysts from fibroadenomas, but not cancers from fibroadenomas (Table 4). The filtered properties, however, significantly improved this mass discrimination, particularly for smaller cancers (Table 4, Figure 5), resulting in statistically significant differences between all mass types. This result can be attributed to the improved contrast for cancer resulting from suppression of stiffness data from adjacent larger benign parenchymal structures whose stiffer properties may have partially obscured and/or diluted the intrinsic stiffness contrast of small masses (Figure 4). However, small cancers also showed an increased percentage of hard components after spatial filtering, particularly in the center, and may be the UST correlate of greater central density noted on mammography for some smaller cancers.

In the case of the larger mass group, the situation was reversed. The spatially filtered data did not yield statistical significance for cancers versus fibroadenomas but the unfiltered data did. This result can also be attributed to the action of the spatial filter which, in addition to suppressing the image rendering of stiffness data from adjacent larger fibroglandular structures (i.e., >1.5 cm), also suppressed the stiffness data within the larger cancers, thereby diluting their intrinsic contrast. In other words, larger cancers don’t require spatial filtering to improve their already high relative image contrast, such that the spatial filter removed ~15% hard components and often left remaining hard components within the margins of larger cancers. It is uncertain whether spatial filtering of large cancers may thus represent an imaging correlate of the fibrotic peri-tumoral region surrounding some larger cancers. In any case, it can be concluded that cancer stiffness appears to decrease with increasing size. A similar but weaker trend is apparent for the benign masses in Figure 5.

The differences noted between smaller and larger masses and the filtered and un-filtered mass images can be explained in part by the physics of the UST imaging process. In the case of the non-filtered cancer images, we note a trend of decreasing stiffness with increasing mass size (Figure 5). This trend can be modeled by understanding how acoustic waves interact with a tumor at various stages of growth. UST captures both the internal distribution of stiffness as well as the overall averaged stiffness of the tumor. For a small tumor, the longitudinal compression wave interacts mainly with the concentrated central tissue of the tumor, resulting in both high sound speed and high attenuation (Figure 6b or Figure 6c, top row). For larger tumors, which may have lower relative collagen deposition, detailed histologic comparisons are needed to better define lower overall stiffness, and may also help confirm any residual hard margins or residual peri-tumoral reactions (24) in larger cancers, but are beyond the scope of this paper.

### 4.3. Current and Future Directions

Our study demonstrated the independence of stiffness from sound speed and adds new information to the interpretation of the whole breast and tissue properties of common masses. Previous work has shown strong correlations between the higher values of whole-breast average UST sound speed and mammographic density, which are representative of the underlying fibroglandular tissue [6,10,12]. In this study we have shown that higher sound speed fibroglandular tissue can be further characterized by its relative stiffness properties. By demonstrating the existence of softer vs. hard fibroglandular tissue by UST, it may be possible to better stratify risk and improve diagnostic accuracy beyond sound speed alone. Therefore, stiffness will likely supplement current sound speed monitoring of risk reduction efforts by tamoxifen chemoprevention [17].

The three common breast masses also showed quantitative differences that provide insight to their visual appearances. These will continue to be refined for diagnostic characterization, but also apply to further evaluation of masses, such as cancer response to neo-adjuvant chemotherapy [13]. SoftVue stiffness imaging thus has many of the benefits of standard US for frequent monitoring of any intervention, including similar projected cost, no radiation, no contrast injection, shorter exam times and appropriateness for younger or pregnant patients.

Weaknesses: a limitation of the stiffness measurement used in this study is the use of a surrogate instead of the actual bulk modulus. Consequently, the stiffness measurement was not calibrated against an external standard and expressed in absolute units. Nevertheless, the bulk modulus surrogate represents a relative estimate of stiffness that can be quantitatively compared between patient groups and may mitigate current elastographic artifacts noted with cysts [3]. Additionally, the component tissue-type distributions within masses only used a single slice surface area ROI to extrapolate their relatively small volumes.

For future screening uses, thorough evaluation of post-processing techniques for whole breast stiffness imaging, such as spatial filtering, is needed before mass characterization benefits can be translated from this initial clinical series to screening populations or studies of mass detection. SoftVue stiffness analyses that need further detail include: comparisons with quantified phantoms, detailed histologic correlates from resection studies, 2D elastographic comparisons with SoftVue, performance differences with post-processing techniques, and stiffness threshold adjustments to its sound speed and attenuation components. Larger studies of SoftVue stiffness imaging may better characterize cancer subtypes and benign findings. While it is encouraging that UST shows promise in ongoing studies of risk evaluation [12], chemoprevention monitoring [17] and imaging responses to neoadjuvant chemotherapy [13], further studies are needed to assess the role of UST stiffness imaging in routine clinical practice. Development of computer-aided diagnostics will automate the labor-intensive, hand-tracing of tumor margins used as a baseline in this study, which will extend to multiple slices for improved volumetric accuracy. Finally, the quantitative nature of SoftVue volumetrics and the textures of stiffness depictions require further analyses of radiomic features with histopathologic correlations.

## 5. Conclusions

We used ultrasound tomography (UST) to volumetrically map stiffness differences in whole breast tissues and common masses. 90% of women had mammographically dense breasts but only 11% of total breast volumes showed hard components, while 69% of fibroglandular tissues were softer. All small cancers (<1.5 cm) showed greater percentage of hard components compared to large cancers (*p* < 0.001). Cancers had greater mean stiffness indices and lower mean homogeneity of stiffness than benign masses (*p* < 0.05). A common finding was the presence of small stiffer masses, mainly due to cancers, amongst predominantly soft breast tissues. SoftVue stiffness imaging using a surrogate of bulk modulus may become a valuable tool for breast cancer detection, risk factor assessments and/or monitoring of chemoprevention or neoadjuvant chemotherapy treatments.

## Figures and Tables

**Figure 1 jcm-10-05528-f001:**
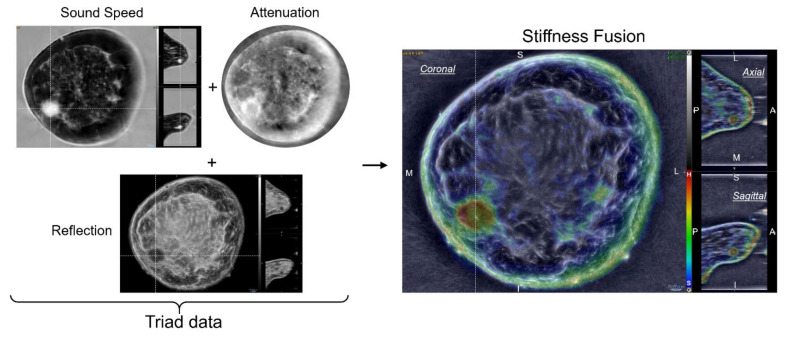
Ultrasound tomography (UST) stiffness imaging is generated from a fusion of a data triad, comprised of quantitative transmission properties from sound speed and attenuation, then overlaid upon co-registered reflection images. Clinical images of a left breast cancer in the 8:00 position (i.e., sound speed, reflection and stiffness fusion) allow further 3D localization by placing the cursor over a suspected mass on the higher resolution coronal images (0.75 mm). The anterior/posterior locations of the stiff cancer (red) can then be viewed in the lower resolution axial and sagittal reconstructions to the right of each coronal image (i.e., 2.5 mm coronal slice thickness). Attenuation is not shown with similar localization since it is only displayed as a component of Stiffness Fusion. Abbreviations: M = medial, L = lateral, S = superior, I = inferior, P = posterior, A = anterior; color scale denotes red as hard (H) and blue as soft (S).

**Figure 2 jcm-10-05528-f002:**
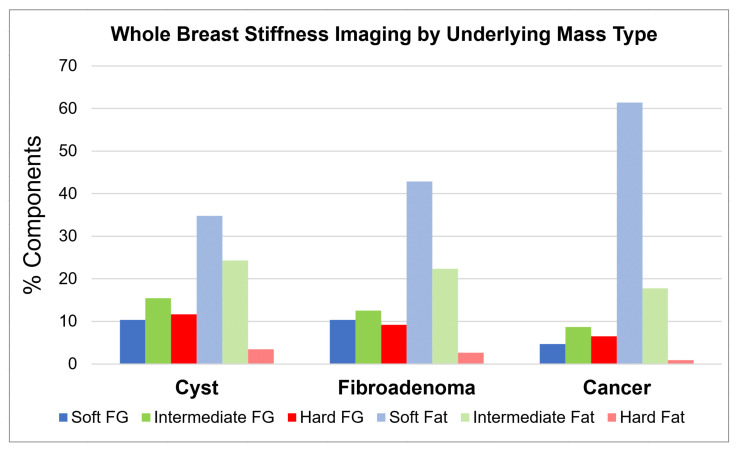
The six whole-breast stiffness components are separated into the three main patient groups according to their underlying masses. Fibroglandular tissues (darker bars) had greater percentages of the stiffest component (deeper red), while fatty tissues had softer components (lighter blue and green bars). SoftVue volumetrics also suggest overall greater percentage of fat for the large majority of patients with dense breasts in this series than by 2D assumptions, similar to volumetric assessments of mammographic breast density (e.g., Volpara, Volpara Health, Lynnwood, WA, USA). This may particularly affect the fat distribution of cancer patients since only patients with scattered breast density (i.e., more fat) had cancerous masses.

**Figure 3 jcm-10-05528-f003:**
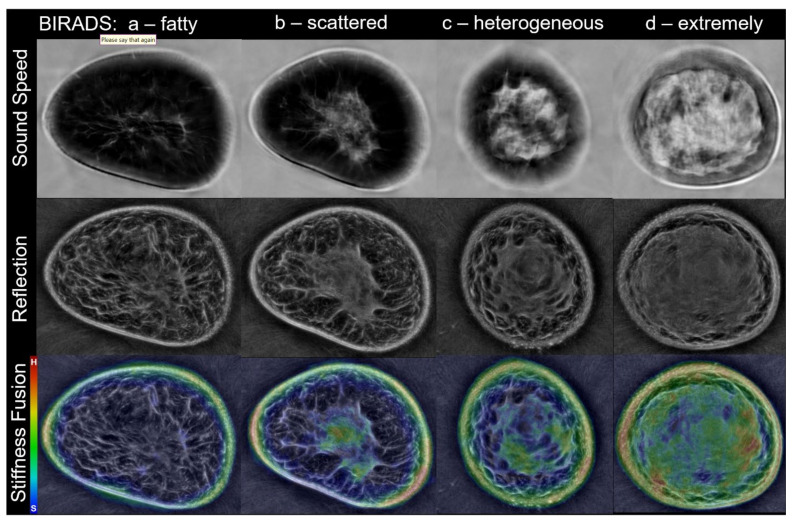
Whole breast stiffness evaluation reveals that most fibroglandular tissue is soft, and the stiffest tissues are a small percentage of total breast volume. Top row: Sound speed images show the gradual increase of fibroglandular tissues (white) from their mammographic density correlates of fatty (left), to extremely dense (right). Middle row: Reflection images show darker fat with increasing proportions of brighter fibroglandular tissue. Bottom row: Corresponding unfiltered stiffness images show their distribution, with blue/black representing the softest tissues and stiffness increasing from green/yellow (intermediate) to red (stiffest). Skin can show artifactual stiffness due to the attenuation component and/or refraction.

**Figure 4 jcm-10-05528-f004:**
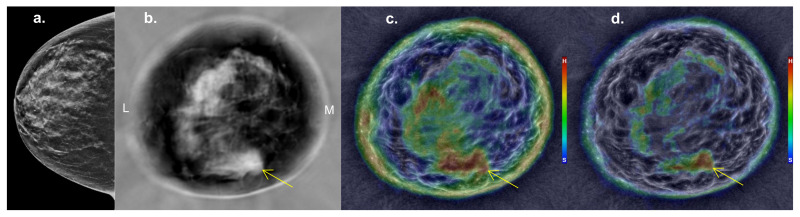
*Cancer patient*: Breast imaging from a woman with heterogeneously dense breasts showing her unremarkable right mammogram in cranio-caudal (CC) projection (**a**), mid-breast sound speed image (**b**), unfiltered (**c**) and filtered (**d**) stiffness images. Mass localization from SoftVue images is similar to coronal MR, whereby the upper portion of the image is superior (e.g., 12:00) and medial (M) for this right breast mass, also defined by the ~5:00 position used in standard US. The unfiltered stiffness image (**c**) shows a larger red area at 5–6:00 that partially obscures the underlying mass effect from the cancer (arrows) at 5:00, better seen on sound speed (**b**) and *filtered* stiffness (**d**) images. Abbreviations: On color bar in part C and D, H = hard, S = soft; italics *filtered* = non-standard stiffness fusion version.

**Figure 5 jcm-10-05528-f005:**
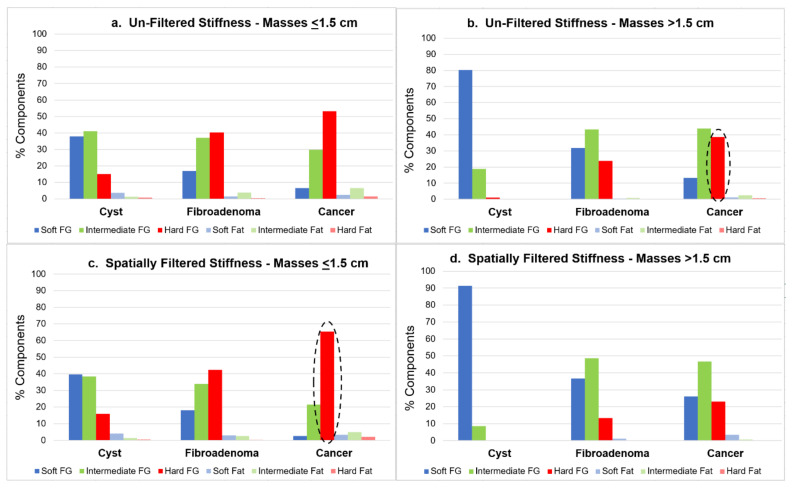
(**a**–**d**)-Graphic distribution of the six relative stiffness components (%) for masses ≤1.5 cm (**a**/**c**-left) and >1.5 cm (**b**/**d**-right), using un-filtered (top row) and spatially filtered (bottom row) renderings (darker bars = fibroglandular = FG; lighter bars = Fat). Smaller masses show greater relative volume of hard components (red) than larger masses, and greater hardness for cancers than fibroadenomas when using spatial filtering (**c**; dashed oval). Conversely, larger masses show greater hard components for the default unfiltered stiffness images, especially cancers (**b**; dashed oval).

**Figure 6 jcm-10-05528-f006:**
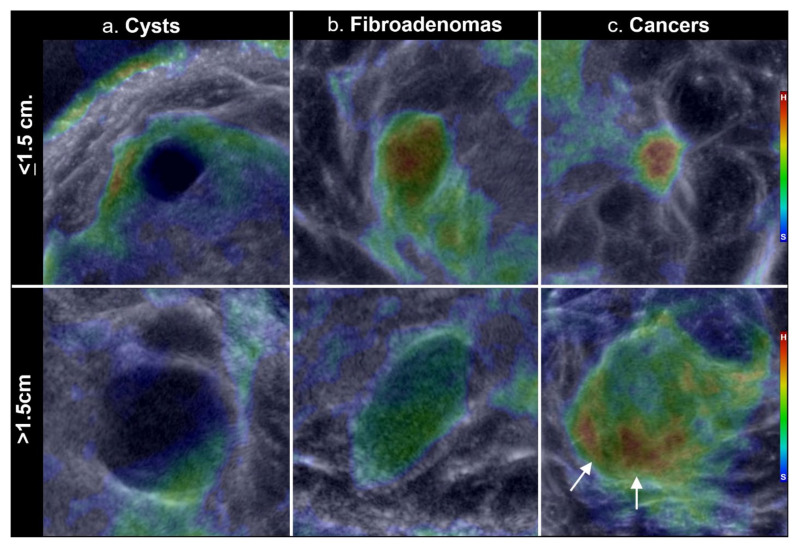
(**a**–**c**)–Filtered stiffness images in columns of cysts (**a**), fibroadenomas (**b**), and cancers (**c**), shown cropped to scale, with the top row showing masses ≤1.5 cm while the bottom row shows comparable type masses >1.5 cm. Note the more blended appearance of a small stiff fibroadenoma (**b**), compared with the more centrally dominant stiffness of a small cancer. The larger invasive ductal carcinoma (C bottom row) was overall softer but had a relatively hard inferior margin (arrows in (**c**), bottom row).

**Table 1 jcm-10-05528-t001:** Current clinical operating parameters of SoftVue UST [5,6,7,12,13,14,15,16,17,18].

SoftVue UST Operating Parameters	
Number of transducer elements	2048
Maximum breast diameter	22 cm
Anatomic coverage (visualized)	Pectoralis muscle to nipple
Operating frequency	3 MHz
Imaging resolution (Superior-inferior × Transverse × Anterior-posterior)	0.75 × 0.75 × 2.5 mm
Data acquisition time per breast	~2 min
Reconstruction time per slice	4 s
Patient throughput (projected)	4 h
Radiologists review time (~complexity)	2–4 min
#Slices per stack (~breast size)	~30–60

**Table 2 jcm-10-05528-t002:** Mass type and size distributions, including subtypes (unbolded) of cancer and other benign categories as noted. (IDC = invasive ductal carcinoma; DCIS = ductal carcinoma in situ; other = 1 mammary, 1 mucinous,1 papillary carcinoma; ILC: invasive lobular carcinoma). The smaller (i.e., <1.5 cm) other benign masses commonly showed underlying fibrosis, noted from biopsy reports (i.e., 13% (21/161) of solid masses).

Mass Histology	Count (N)	<1.5 cm	>1.5 cm
**Cancer**	**78**	**52**	**26**
Subtypes: IDC	57	37	20
DCIS alone	6	5	1
ILC	10	5	5
Other	3	3	0
DCIS + IDC	2	2	0
**Fibroadenoma**	**105**	**88**	**17**
**Cyst**	**91**	**80**	**11**
**Other benign**	**24**	**21**	**3**
Subtypes: Containing fibrosis		21	
Fibrocystic change			2
Granulomatous Mastitis			1
**Totals**	**298**	**241**	**57**

Bold: the major histology types.

**Table 3 jcm-10-05528-t003:** The average volumetric distributions of the three stiffness components for fibroglandular and fatty tissues throughout the average whole breast (i.e., mean total breast volume = 737 cc; N = 239 breasts).

	Fibroglandular		Fatty		Total	
	(cc)	% Total	(cc)	% Total	(cc)	% Total
**Hard**	66	9.0%	16	2.2%	82	**11.2%**
**Intermediate**	87	11.8%	160	21.7%	247	**33.4%**
**Soft**	61	8.3%	347	47.1%	408	**55.4%**
**Total**	214	**29.1%**	523	**71.0%**	737	**100.0%**

Bold denotes major stiffness and tissue categories, and the associated total percentages.

**Table 4 jcm-10-05528-t004:** Mass comparisons according to size and stiffness values using unfiltered and filtered stiffness values (also italicized). The spatially filtered algorithm accentuated smaller masses, producing significantly better differentiation of cancers from fibroadenomas (bolded). Conversely, spatial filtering degraded performance for larger, solid mass differentiation (bolded).

Total Mass Comparisons			Stiffness (5–95% C.I)		*p* Values
**Small**	**Unfiltered**	Cancer (CA)	0.1256–0.8585	CA vs. Cyst:	0.000000001
(≤1.5 cm)		Fibroadenoma (FA)	0.0021–0.8634	**CA vs. FA:**	**0.08**
		Cyst	0.00012–0.6499	Cyst vs. FA:	0.0000017
	** *Filtered* **	*Cancer (CA)*	*0.1247–0.6967*	*CA vs. Cyst:*	*6 × 10^−11^*
		*Fibroadenoma (FA)*	*0.1704–0.5303*	** *CA vs. FA:* **	** *0.000036* **
		*Cyst*	*0.001–0.3618*	*Cyst vs. FA:*	*0.00028*
**Large**	**Unfiltered**	Cancer (CA)	0.1916–0.8288	CA vs. Cyst:	*2 × 10^−10^*
(>1.5 cm)		Fibroadenoma (FA)	0.036–0.7005	**CA vs. FA:**	**0.037**
		Cyst	0.00034–0.2364	Cyst vs. FA:	0.00013
	** *Filtered* **	*Cancer (CA)*	*0.0826–0.8104*	*CA vs. Cyst:*	*0.00000021*
		*Fibroadenoma (FA)*	*0.0197–0.4830*	** *CA vs. FA:* **	** *0.127* **
		*Cyst*	*0.00005–0.0691*	*Cyst vs. FA:*	*0.000025*

## Data Availability

Restrictions apply to the availability of these data. Data were obtained from Delphinus Medical Technologies, Inc. (DMT) and have limited availability from the authors with the permission of DMT.

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
