# Peer review of "Multicenter Study of Whole Breast Stiffness Imaging by Ultrasound Tomography (SoftVue) for Characterization of Breast Tissues and Masses"

_jcm, 2021, doi:10.3390/jcm10235528_

Round 1
Reviewer 1 Report
This study presents a new technique using a combination of sound speed and attenuation data acquired on a ultrasound tomography machine, as a surrogate of breast "stiffness", different from the usual shear wave or strain techniques.
The manuscript provides very innovative insights on the description and potential advantages of this new technique. Interestingly, the work includes both a "breast-wise" and "lesion-wise" analysis, representing promising avenues of research. Imaging presented in this manuscript is very appealing.
However for now it is not clear which are the promoted clinical applications of these technique : the manuscript largely deals with lesion-wise stiffness and differences in stiffness between cyst/FA/tumors, while the conclusion (« current and future directions ») highlights the use of breast-wise stiffness to stratify breast cancer risk or evaluate response to chemoprevention.
Lesion-wise stiffness is analyzed in term of the relative presence of 6 composite entities : « soft » « intermediate » and « hard » fibroglandular tissue, « soft » « intermediate » and « hard » fat.
It is not clear how the partition between « soft » « intermediate » and « hard » was done, given the authors recognize that the stiffness parameter is not expressed in absolute units.
There is no assessment of the reproducibility of the stiffness.
I would be interested in knowing how much of the « hard » tissue -as defined by K-means clustering or the « red » color - is indeed malignant. Additionally, how does the technique behave in dense breasts, where neighboring normal tissue is hard ?
From figure 5 it seems that stiffness value of fibroadenoma and cancer are somewhat close. Classification seems to be improved, for small lesions, by using a spatial filter. How this filter affects detection of lesions ?
A portion of benign lesions were seemingly excluded from the analysis (table 2 : other benign, 24 lesions), while they could represent a significant proportion of false positive of this technique.
Line 77 : how the size cutoff of 1.5 cm was chosen ?
Line 135 : the sum of all six regions
Line 172 : were the patients with scattered breast density added to the initial population of the larger study SV PCC ARM2 ? If so, how were these patients selected ?
Author Response
Reviewer #1
Thank you for the constructive comments that we have hopefully answered to your satisfaction and made this a better manuscript. A recent major event occurred from the time of this manuscript submission until now – the FDA Pre-Market Approval of SoftVue as an adjunct to screening mammography in women with dense breasts. We only mentioned it briefly in the article but the PMA further highlights the clinical relevance of this paper, which may now may become a realistic framework for further ongoing work in this field.
We have numbered the following comments and/or questions for each reviewer. The line number locations of all corrections are noted from the “accept changes” version of the manuscript, but a “tracked changes” version is also submitted for easier tracking of location changes.
#1. “it is not clear which are the promoted clinical applications of these technique : the manuscript largely deals with lesion-wise stiffness and differences in stiffness between cyst/FA/tumors, while the conclusion (« current and future directions ») highlights the use of breast-wise stiffness to stratify breast cancer risk or evaluate response to chemoprevention.”
The reviewer is correct in noting that this paper deals with both lesion-wise and breast-wise stiffness evaluations, but it does NOT relate to screening and "detection". We have therefore made the following changes to more clearly emphasize the whole-breast and mass specific aspects. Two sentences were added to the Introduction (i.e., lines 53-54, and 60-61) to address the need for both whole-breast and mass-specific diagnostic implications of stiffness visualization. The Discussion section also added 2 sentences (i.e., lines 312-14, and 330-2) emphasizing the whole-breast and mass-specific data. The Current and Future Directions section has also been separated into a whole-breast and mass-specific paragraphs for clarity.
#2. "It is not clear how the partition between « soft » « intermediate » and « hard » was done, given the authors recognize that the stiffness parameter is not expressed in absolute units.
There is no assessment of the reproducibility of the stiffness.”
The surrogate bulk modulus is expressed in quantified pixel values that are consistently obtained and reproducible from patient to patient as noted in the third paragraph of Section 2.2 (lines 116-125). In addition, we hopefully further clarified their application in Section 2.3 using K-means clustering (lines 125-135). Since both Sound Speed and Stiffness use quantified pixel values they can both be readily separated by K-means clustering, regardless whether they have absolute units (i.e., m/sec) or not, respectively.
#3. I would be interested in knowing how much of the « hard » tissue -as defined by K-means clustering or the « red » color - is indeed malignant. Additionally, how does the technique behave in dense breasts, where neighboring normal tissue is hard ?
This is an interesting question that we have clarified in several ways. We added a paragraph to Discussion Section 4.1 Whole Breast Stiffness (lines 341-50) to suggest that stiffness visualization of underlying masses is a volumetric balance of relative stiffness. Namely, how do masses appear upon a background of whole-breast stiffness, particularly within the 90% of women with dense breasts in this paper? As noted in Table 3, there is only ~9% average breast volume, or 66cc, showing hard fibroglandular tissue scattered throughout the average 737mL breast. The focal accumulation of these 66cc will determine how much they could obscure the average hard 1.1mL mass. Therefore, we have also included several sentences that also emphasize the importance of clinically reviewing Stiffness Fusion images as the last step in mass characterization (M&M lines 114-5; Results lines 270-76). Hard parenchyma is differentiated as its colors “flow” while scrolling through stiffness images, whereas stiffness colors appear to “stick” to any true underlying masses (Results lines 270-76).
#4. From figure 5 it seems that stiffness value of fibroadenoma and cancer are somewhat close. Classification seems to be improved, for small lesions, by using a spatial filter. How this filter affects detection of lesions ?
As noted above in #1, this paper is limited to a clinical series of known masses, exploring their diagnostic characterization by Stiffness Fusion imaging. We hoped that it would represent a framework for future additional work, thereby only representing the first step towards eventual application to screening, or better detection of underlying masses. If this wasn’t clear from the initial manuscript, we have re-emphasized this in several locations in the Discussion: lines 312-14; 330-22; and 345-50.
#5. A portion of benign lesions were seemingly excluded from the analysis (table 2 : other benign, 24 lesions), while they could represent a significant proportion of false positive of this technique.
These “other benign” masses were NOT excluded from analyses, just their representation in Figures 2&5, in order to simplify those figures. They have been more directly emphasized as false positives, or 13% of solid masses in Table 2 (line 189), then in lines 282-86.
#6. Line 77 : how the size cutoff of 1.5 cm was chosen ?
While it could be considered somewhat arbitrary, the 1.5cm threshold was a combination of the spatial filter criterion and the common clinical size target for screening masses, in the range of 5-15mm, which may be commonly missed by mammography. This is noted in lines 81-3.
#7. Line 135 : the sum of all six regions
Corrected missing “of” in line 133.
#8. Line 172 : were the patients with scattered breast density added to the initial population of the larger study SV PCC ARM2 ? If so, how were these patients selected ?
No, they were already part of the clinical group of patients in ARM2 selected for an underlying mass being worked up for biopsy, similar to all the other patients with masses. This is more clearly noted in lines 176-77.

Reviewer 2 Report
This study uses a parameter called stiffness estimated by UST images as a metric to differentiate between cysts, cancer, and fibroadenomas in the breast.
The study of new metrics and image features for classification of findings in UST is important and can further enhance the clinical value of this tool. The proposed parameter, "bulk modulus surrogate" as it is called by the authors, may assist in the classification of the three types of breast lesions studied in this work, Cyst, CA, and FA. There are three major concerns in an otherwise well executed study and well written paper:
(1) The results in Table 4 are not impressive. The metric works well in the differentiation between CA and Cyst or Cyst and FA, for either small or large filtered masses. It works well for the differentiation of CA and FA in the filtered small masses but fails in the unfiltered images as it does for the large masses. The explanation of this result is attempted in lines 267-272 but it not clear why the hard components within the rim of the large masses would make such a different. Why does the location of the hard components affect the outcome as presented in Figure 1? Please explain in more detail. The lack of a good understanding of this result may have a significant impact on the clinical value (if any) of the proposed stiffness parameter.
2) Related to the previous issue may be the fact that manual outlines of the masses were generated by experts and these outlines were used for the stiffness measurements. If the rim plays such an important role on the outcome, then the manual outlines should add to the uncertainly of the estimate. Do small differences in manual outlines affect the results and how? Have you tried to reproduce the measurements with outlines of the same masses from different expert?
3) A high pass spatial filter is commonly used for postprocessing of images. As such, it may not have been the best way to evaluate the proposed stiffness parameter because it adds another variable to the evaluation and makes it difficult to assess the true value of the proposed metric.
Finally, please clarify whether the "stiffness parameter" is the result of attenuation and speed info as stated in line 117 or attenuation, speed, and reflection as shown in Fig. 1.
Author Response
Reviewer #2
Thank you for the constructive comments that we have hopefully answered to your satisfaction and made this a better manuscript. A recent major event occurred from the time of this manuscript submission until now – the FDA Pre-Market Approval of SoftVue as an adjunct to screening mammography in women with dense breasts. We only mentioned it briefly in the article but the PMA further highlights the clinical relevance of this paper, which may now may become a realistic framework for further ongoing work in this field.
We have numbered the following comments and/or questions for each reviewer. The line number locations of all corrections are noted from the “accept changes” version of the manuscript, but a “tracked changes” version is also submitted for easier tracking of location changes.
There are three major concerns in an otherwise well executed study and well written paper:
(1) The results in Table 4 are not impressive. The metric works well in the differentiation between CA and Cyst or Cyst and FA, for either small or large filtered masses. It works well for the differentiation of CA and FA in the filtered small masses but fails in the unfiltered images as it does for the large masses. The explanation of this result is attempted in lines 267-272 but it not clear why the hard components within the rim of the large masses would make such a different. Why does the location of the hard components affect the outcome as presented in Figure 1? Please explain in more detail. The lack of a good understanding of this result may have a significant impact on the clinical value (if any) of the proposed stiffness parameter.
The reviewer is correct in the lack of continuity between Table 4 and the more qualitative appearance of masses in Figure 6. While Table 4 may not appear impressive, it validates the role of a spatial filter improving discrimination of all smaller masses, while decreasing differentiation of larger masses since it downgrades some of the harder components. We agree this needed greater clarity, so we first expanded the Results description of Figure 6 in relation to their image interpretation in lines 270-76. The cancer-specific paragraph that followed later was also updated to emphasize that the filter actually makes small cancers appear ~23% harder, or more red, while making larger cancers appear ~15% less red (lines 287-97; 359-60). This may be a reason that filtered images of larger masses thus may leave only a margin that has residual hardness (lines 292-3, 365-7). Whether this “hard rim” relates to any underlying greater fibrous reaction at some margins of larger tumors is postulated in lines 368-9, requiring further histologic comparisons, noted in 377-9 and 401-404.
2) Related to the previous issue may be the fact that manual outlines of the masses were generated by experts and these outlines were used for the stiffness measurements. If the rim plays such an important role on the outcome, then the manual outlines should add to the uncertainly of the estimate. Do small differences in manual outlines affect the results and how? Have you tried to reproduce the measurements with outlines of the same masses from different expert?
Thank you for pointing this out since we also didn’t adequately describe that these manual outlines of tumor margin initially also included an automated peri-tumoral region surrounding each mass tracing. A recent paper had already validated and described quantitative peri-tumoral analyses, which confirmed that >90% of cancers occur at the fat-glandular interface (now ref#24). We moved this reference up to address its prior use, also emphasizing the low probability of any significant margin tracing differences, which were only done on the best visualized combination of Sound Speed and Reflection images. As with the prior study [24], we believe that minimal differences in margin tracing between breast imagers would produce any significant differences since automated solutions for mass margin detection are also progressing that would provide more volumetric assessments using multiple slices. Additional sentences were thus added to M&M Section 2.4, lines 143-7. But it does remain a weakness that we already noted in Discussion lines 397-8; 407-11.
3) A high pass spatial filter is commonly used for postprocessing of images. As such, it may not have been the best way to evaluate the proposed stiffness parameter because it adds another variable to the evaluation and makes it difficult to assess the true value of the proposed metric.
The reviewer is correct in that we only presented the use of a spatial filter as an exploratory exercise in post-processing to see whether qualitative differences in masses could be better quantified and produce better mass discriminations, particularly for masses in the common screening range of 5-15 mm. Both Stiff Fusion presentations are applicable, but spatial filtering should not yet be used on whole-breast evaluations since that enters the realm of mass detection for screening and the need to carefully assess for associated false positives, etc. We hoped that application of at least a spatial filter as the only post-processing done to date, in order to improve SoftVue mass differentiation could provide a framework for future work. If this wasn’t clear from the initial manuscript, we have re-emphasized this in several locations in the Discussion: lines 345-50, 357-60, 362-5; as well as still noting it as a potential weakness needing more thorough assessment of “performance differences with postprocessing techniques “ in line 400.
4) Finally, please clarify whether the "stiffness parameter" is the result of attenuation and speed info as stated in line 117 or attenuation, speed, and reflection as shown in Fig. 1.
The Reviewer is correct that the “stiffness parameter” combines attenuation and sound speed data as their product, which eliminates any absolute physical units (i.e., m/sec * dB/MHz/cm), but still produces pixels with reproducible quantitative values. These pixels are displayed on a relative color scale, then overlaid upon a background of the Reflection image at that level. In order words, no Reflection value is included in the quantitative stiffness parameter. As noted for the other Reviewer, the third paragraph of Section 2.2 (lines 116-24) described the concept of a surrogate bulk modulus still retaining quantified pixel values of relative stiffness that are consistently obtained and reproducible from patient to patient, as well as readily separable on a relative stiffness scale. In addition, we hopefully further clarified their application in Section 2.3 using K-means clustering (lines 125-135).

Round 2
Reviewer 1 Report
My remarks have been correctly addressed.
I have no further concerns.